# Efficacy of Adaptogens in Patients with Long COVID-19: A Randomized, Quadruple-Blind, Placebo-Controlled Trial

**DOI:** 10.3390/ph15030345

**Published:** 2022-03-11

**Authors:** Irina Karosanidze, Ushangi Kiladze, Nino Kirtadze, Mikhail Giorgadze, Nana Amashukeli, Nino Parulava, Neli Iluridze, Nana Kikabidze, Nana Gudavadze, Lali Gelashvili, Vazha Koberidze, Eka Gigashvili, Natela Jajanidze, Naira Latsabidze, Nato Mamageishvili, Ramaz Shengelia, Areg Hovhannisyan, Alexander Panossian

**Affiliations:** 1National Family Medicine Training Centre, 57 M. Tsinamdzghvrishvili Str., Tbilisi 0102, Georgia; nfmtc@nilc.org.ge (I.K.); ukakiladze@gmail.com (U.K.); kirtadzenino.nfmtc@gmail.com (N.K.); mgiorgadze.nfmtc@gmail.com (M.G.); nanakeli180818@gmail.com (N.A.); nino.parulava69@mail.ru (N.P.); neliiluridze3.nfmtc@gmail.com (N.I.); nanakikabidze2@gmail.com (N.K.); nanagudavadze@gmail.com (N.G.); laligelashvili777@gmail.com (L.G.); vajakoberidze56@gmail.com (V.K.); gigashvili.nfmtc@gmail.com (E.G.); jajanidze.nfmtc@gmail.com (N.J.); nairalacabidze@gmail.com (N.L.); 2Department for History of Medicine and Bioethics, Faculty of Medicine, Tbilisi State Medical University, Vazha-Pshavela Ave. 33, Tbilisi 0162, Georgia; nato15sg@yahoo.com (N.M.); r.shengelia@tmsu.edu (R.S.); 3Institute of Fine Organic Chemistry, National Academy of Science, Azatutian ave. 26, Yerevan 375014, Armenia; dopingareg@gmail.com; 4Phytomed AB, Bofinksvagen 1, 31275 Våxtorp, Sweden

**Keywords:** adaptogens, Chisan^®^/ADAPT-232, clinical trial, long COVID-19 symptoms, physical performance, IL-6, D-dimer, creatinine, C-reactive protein

## Abstract

Currently, no effective treatment of comorbid complications or COVID-19 long-haulers during convalescence is known. This randomized, quadruple-blind, placebo-controlled trial aimed to assess the efficacy of adaptogens on the recovery of patients with Long COVID symptoms. One hundred patients with confirmed positive SARS-CoV-2 test, discharged from COVID Hotel isolation, Intensive Care Unit (ICU), or Online Clinics, and who experienced at least three of nine Long COVID symptoms (fatigue, headache, respiratory insufficiency, cognitive performance, mood disorders, loss of smell, taste, and hair, sweatiness, cough, pain in joints, muscles, and chest) in the 30 days before randomization were included in the study of the efficacy of Chisan^®^/ADAPT-232 (a fixed combination of adaptogens Rhodiola, Eleutherococcus, and Schisandra) supplementation for two weeks. Chisan^®^ decreased the duration of fatigue and pain for one and two days, respectively, in 50% of patients. The number of patients with lack of fatigue and pain symptoms was significantly less in the Chisan^®^ treatment group than in the placebo group on Days 9 (39% vs. 57%, pain relief, *p* = 0.0019) and 11 (28% vs. 43%, relief of fatigue, * *p* = 0.0157). Significant relief of severity of all Long COVID symptoms over the time of treatment and the follow-up period was observed in both groups of patients, notably decreasing the level of anxiety and depression from mild and moderate to normal, as well as increasing cognitive performance in patients in the d2 test for attention and increasing their physical activity and workout (daily walk time). However, the significant difference between placebo and Chisan^®^ treatment was observed only with a workout (daily walk time) and relieving respiratory insufficiency (cough). A clinical assessment of blood markers of the inflammatory response (C-reactive protein) and blood coagulation (D-dimer) did not reveal any significant difference over time between treatment groups except significantly lower IL-6 in the Chisan^®^ treatment group. Furthermore, a significant difference between the placebo and Chisan^®^ treatment was observed for creatinine: Chisan^®^ significantly decreased blood creatinine compared to the placebo, suggesting prevention of renal failure progression in Long COVID. In this study, we, for the first time, demonstrate that adaptogens can increase physical performance in Long COVID and reduce the duration of fatigue and chronic pain. It also suggests that Chisan^®^/ADAPT-232 might be useful for preventing the progression of renal failure associated with increasing creatinine.

## 1. Introduction

COVID-19, caused by SARS-CoV-2, can involve sequelae and other medical complications that last weeks to months after initial recovery, which has come to be called Long COVID [1,2]. Many COVID-19 patients in convalescence still had symptoms and lung inflammation after hospital discharge and recovered with time, with symptoms prolonging for 2–3 months [1,2,3,4,5]. Symptoms persisting two or more weeks after COVID-19 onset that do not return to a healthy baseline can potentially be considered long-term effects of the disease [3]. A systematic review and meta-analysis in a total of 18,251 publications on 47,910 patients revealed that 80% of the patients developed one or more long-term symptoms including fatigue, headache, attention disorder, hair loss, and dyspnea in the follow-up time ranging from 14 to 110 days post-viral infection [2]. Another systematic review and meta-analysis in a total of 1963 studies on 3559 COVID-19 patients from China, South Korea, Saudi Arabia, France, Japan, Singapore, Canada, the UK, and the USA concluded that SARS-CoV-2 might cause delirium in the acute stage and fatigue, depression, anxiety, post-traumatic stress disorder, and neuropsychiatric syndromes in the longer term [6].

Over 7479 clinical studies in COVID-19, including 276 interventional trials of antiviral treatments in Long COVID-19, are currently in progress [7,8,9,10,11,12,13,14,15,16,17,18,19,20,21,22,23,24,25,26,27,28,29,30,31,32,33,34]. Recent studies show that some Traditional Chinese medicines promote the improvement process and alleviate lung inflammation for early recovery [34]. However, little is known regarding the effective treatment of comorbid complications or COVID long-haulers during convalescence [35,36].

This study aims to assess the efficacy of an adaptogenic preparation, Chisan^®^/ADAPT-232, on recovery of COVID-19 patients in the rehabilitation period after discharge from Intensive Care Units and Online Clinics.

Chisan^®^/ADAPT-232 is a pharmaceutical formulation of a fixed combination of adaptogens Rhodiola, Eleutherococcus, and Schisandra, used in different dosage forms (capsules and the oral solution Chisan^®^) for the enhancement of mental and physical capacities in case of tiredness or during convalescence [37,38,39,40,41]. The active ingredients of Chisan^®^/ADAPT-232 are extracts from *Rhodiola rosea* roots, *Schisandra chinensis* berry, and *Eleutherococcus senticosus* root with adaptogenic, stimulating, and stress-protective activities [40,41,42]. The combination of these ingredients synergistically contributes to the efficacy of Chisan^®^/ADAPT-232. It has been shown that combining these three extracts in the ADAPT-232 combination induces both synergistic and antagonistic interactions of molecular networks of targeted cells, resulting in the deregulation of new genes specific exclusively for the Chisan^®^/ADAPT-232 combination [42]. These interactions may influence transcriptional control of metabolic regulation both in isolated cells and on the higher levels of regulations of the entire organism [41], inducing possible benefits in COVID-19 [42] and during the recovery of patients in respiratory diseases [39]. This proposal is in line with the results of a randomized placebo control study of Chisan^®^/ADAPT-232 in pneumonia [39], where Chisan^®^/ADAPT-232 significantly decreased the duration of antibiotic therapy, reducing the length of the acute phase of the illness, and improving the mental performance and the quality of life of patients.

## 2. Results

### 2.1. Patients

Demographic and Baseline Characteristics

Table 1 shows patients’ baseline demographic and clinical characteristics in both study groups. The groups did not differ in age, weight, body mass index, and other study outcome measures.

### 2.2. Efficacy

The severity of Long COVID inflammatory symptoms gradually decreased from the baseline to the end of therapy (Day 14) and follow-up period (21 days) in both treatment groups. Significant relief of fatigue, headache, respiration insufficiency (dyspnea or polypnea), sweatiness, loss of smell (anosmia) and taste (ageusia), cough, hair loss, pain in joints, muscles, and chest, cognitive functions (attention, memory), mood disorders (anxiety, and depression) and physical activity over time of treatment and the follow-up period was observed in both groups of patients (Figure 1a and Appendix A).

The therapeutic efficacy of ADAPT-232 was assessed by comparing differences in the relief of inflammatory symptoms from the baseline in ADAPT-232 and placebo groups of patients (Figure 1b and Appendix A). Treatment groups were compared for all critical measures of efficacy (primary and secondary endpoints).

Figure 1 shows the effect of ADAPT-232 treatment on physical performance of patients compared to placebo over the time of treatment and follow-up period, while the changes in the level of blood inflammatory (IL-6, C-reactive protein) and coagulation (D-dimer) markers are demonstrated in Figure 2 and Figure 3.

Table 2, Figure 4, and Appendix A show the duration (days) from randomization to when symptoms disappear; for details, see Appendix A.

Figure 4 shows the duration of fatigue and pain over the time from randomization (Day 1) to the end of the treatment (Day 14) and follow-up for one week (Day 21) in patients who experienced these symptoms. ADAPT-232 decreased the duration of fatigue and pain for one and two days, respectively, in 50% of patients. The number of patients with symptoms of fatigue and pain was less in the ADAPT-232 treatment group (39% and 28%, pain and fatigue, correspondingly) than in the placebo group (57% and 43%, pain and fatigue correspondingly). The difference is statistically significant on Day 9, ** *p* = 0.0019—pain relief, and Day 11, * *p* = 0.0157—fatigue.

### 2.3. Safety

Only one adverse event (allergic conjunctivitis) was recorded in the patient study using a placebo. Regardless of causality, treatment-emergent adverse events were monitored for all patients from the first dose and through the one-week follow-up period.

## 3. Discussion

Overall, 105 post-COVID-19 patients with confirmed diagnoses based on a positive SARS-CoV-2 test, discharged from COVID Hotel isolation, Intensive Care Unit (ICU), or Online Clinics, and experiencing COVID symptoms in the 30 days before randomization were assessed for eligibility.

One hundred eligible patients of 48.85 ± 13.9 years old with at least 3 of 9 Long COVID symptoms (fatigue, headache, respiratory insufficiency, cognitive performance, mood disorders, loss of smell, taste, and hair, sweatiness, cough, pain in joints, muscles, and chest) were randomly assigned to two treatment groups, A and B. The groups did not show differences at baseline demographic, physical, and other critical clinical measurements carried out during the study or identified as important indicators of prognosis or response to therapy characteristics.

Substantial and increasing cognitive performance patients in the d2 test for attention and increasing their physical activity and workout (daily walk time) was recorded by objective measures. Statistically significant relief of all tested Long COVID symptom overtreatment and the follow-up period was observed in both groups of patients, including fatigue, chronic pain, and decrease of anxiety and depression from mild to moderate to normal. However, a significant difference between placebo and ADAPT-232 treatment was observed in relief of respiratory insufficiency (cough) and workout (daily walk time). ADAPT-232 significantly increased daily workout compared to placebo (Figure 1). which correlates with a decrease in the duration of fatigue and pain (Figure 2), despite insignificance in the difference of severities of these symptoms in subjective self-assessment test (Appendix A).

ADAPT-232 decreased the duration of fatigue and pain for one and two days, respectively, in 50% of patients, and the number of patients with symptoms of fatigue and pain was less in the ADAPT-232 treatment group than in the placebo group, e.g., this difference was statistically significant on Day 9, ** *p* = 0.0019—pain relief, and Day 11, * *p* = 0.0157—fatigue.

Different cell types in humans secrete Interleukin-6 (IL-6). A meta-analysis on a total of 140 studies, including 12,421 values for IL-6 in the blood of healthy adult donors, reported that the values for IL-6 in the blood of healthy donors varied between 0 and 43.5 pg/mL. The pooled estimate of IL-6 was 5.186 pg/mL (95% confidence interval (CI): 4.631, 5.740), increasing with age by 0.05 pg/mL/year (95% CI: 0.02, 0.09; *p* < 0.01) [43]. IL-6 is associated with different diseases and viral infections, including COVID-19, where the median concentration of IL-6 gradually increases from 1.5 pg/mL to 21.55 pg/mL depending on the severity of inflammatory symptoms [44]. The concentration of IL-6 > 24 pg/mL predicted the development of hypoxemia [45], while the concentration higher than 37.65 pg/mL was predictive of in-hospital death [44].

In post-acute COVID-19, the concentration of serum IL-6 was found to be 1.45 ± 2.1 pg/mL, 1.96 ± 1.9 pg/mL, and 4.43 ± 6.6 pg/mL in patients with anamnesis of mild, moderate, and severe COVID-19, respectively, where the severity of the disease was characterized according to the need of medical treatment: mild, outward treatment; moderate, inward treatment; severe, inward treatment respiratory support (oxygen supply or mechanical ventilation) [46].

In our study, the baseline level of serum IL-6 mean was found to be 6.443 ± 1.548 pg/mL and 5.883 ± 0.554 pg/mL in *verum* and placebo groups. After two weeks of treatment with ADAPT-232/Chisan^®^ and one week of follow-up, at the end of the study, serum concentration of IL-6 significantly decreased to 4.730 ± 0.329 pg/mL (difference—1.2508; *p =* 0.0487, multiple *t*-tests), while in the placebo group, it increased insignificantly to 5.981 ± 0.530 pg/mL.

Recently, a new clinical marker of lung injury severity and progression of COVID-19 was found, Neuron-Specific Enolase (NSE), which is localized in the cytoplasm of neurons and neuroendocrine cells [47]. Further studies of interest were the correlation between serum NSE level in the acute phase of COVID patients (who were referred to the emergency department or Intensive Care Unit) and the effects of adaptogens in the same patients in the post-COVID phase.

Clinical assessment of other blood markers of the inflammatory response (C-reactive protein), blood coagulation (D-dimer), and renal failure (creatinine) did not reveal any significant changes over time between the treatment groups (Figure 2). Their level was within the normal limits; however, a significant difference between the placebo and ADAPT-232 treatment was detected for creatinine: ADAPT-232 significantly decreased blood creatinine compared to placebo, suggesting prevention of renal failure progression in Long COVID.

Serum creatinine derived from phosphocreatine during energy-producing processes in muscles, brain, and myocardium is effectively eliminated in kidneys from the blood circulation system. An abnormally elevated creatinine level in blood indicates poor clearance of creatinine, malfunction of kidneys, and possible progression of kidney disease due to insufficient detoxification of the organism in acute or chronic inflammation [48,49,50,51].

Kidney dysfunction is common in COVID patients and may result in a progressive decline in kidney function and chronic kidney disease [50]. It was reported that patients who survived coronavirus disease 2019 are at higher risk of post-acute sequelae, including pulmonary and kidney disease [48]; 14.4% of patients with COVID-19 had elevated serum creatinine, and high discharge serum creatinine was associated with non-recovery of kidney disease [51]. In this context, the results of our study also suggest that ADAPT-232 might be useful for the prevention of renal dysfunction in post-COVID patients, particularly in elderly subjects at high risk for development of kidney dysfunction [51].

There is one study regarding Chinese Medicine (CM) where two herbal teas containing (a) the combinations of 10 antiviral herbal substances including Salvia miltiorrhiza, Fructus hordei germinates, Fructus hordei germinates, Codonopsis pilosula, Adenophora stricta, Peach kernel, Melon burdock, Magnolia officinalis, Radix Reed, and Herba patriniae, and (b) the combinations of five tonic herbal substances including Radix adenophorae, Ophiopogon japonicus, Astragalus membranaceus, Rhizoma Dioscoreae, and Massa Fermentata were studied in 96 patients with coronavirus disease 2019 (COVID-19) who had not recovered after hospital discharge. There was a prospective cohort and nested case-control open-label study in 64 patients who received 28-day CM treatment and 32 patients comprising the control group without CM [34]. There was no significant difference between the two groups in the improvement rates of respiratory and other symptoms, including fatigue, sputum, cough, dry throat, thirst, and upset stomach. However, on Day 14, the CM group had a significantly higher absorption rate than the control group, suggesting that CM treatment could improve lung inflammation [34].

On the contrary, our study, conducted with a pharmaceutical-grade herbal medicinal product according to ICH guidelines for good clinical practice, suggests that Chisan^®^/ADAPT-232, a fixed combination of adaptogens, can be helpful in the recovery of post-COVID patients who experienced decreased physical activity, fatigue, chest, joint and muscle pain, and high pro-inflammatory markers IL-6 and creatinine.

This study provides new clinical evidence on the efficacy of adaptogens and specifically ADAPT-232 during the recovery of patients after viral infection.

The results of this study are also consistent with our previous observations on the beneficial effect of ADAPT-232 on the recovery of patients with pneumonia [37], as well as with the results of other studies of adaptogens where the mechanisms of detoxification and reparation of oxidative stress-induced damages in compromised cells, cytoprotective, antioxidant and repairing mechanisms of actions were elucidated [42]. They are associated with activation of adaptive signaling pathways [41], activation of Nrf2-mediated oxidative stress response signaling pathway proteins (KEAP1), production of Phases I and Il metabolizing and antioxidant enzymes, reduction of oxidative stress-induced molecular damages, and molecular chaperon Hsp70-mediated cytoprotective, detoxifying and repairing processes and cellular dysfunctions [41,42,52].

Finally, this study provides clinical evidence of stimulating (physical activity) and stress-protective (preventing viral infection-induced nephrotoxicity) effects of ADAPT-232, the most specific characteristics of adaptogens.

## 4. Materials and Methods

### 4.1. Study Design

Recruitment for the study was initiated on 24th April 2021, and the last person was recruited on 22nd October 2021. Overall, 105 patients were assessed for eligibility, and 100 patients (48.85 ± 13.9 years old) with Long COVID symptoms were in the study (Appendix A, Full Analysis Dataset). In total, 99 patients (99%) completed their respective treatment cycles according to protocol, while one patient (1%) discontinued therapy after receiving at least one dose of trial medications due to the patient’s request. Ninety-nine patients, who completed the treatment, were evaluated for treatment efficacy for three weeks (visits 2 and 3) of treatment and one week after finishing the treatment (visit 4, follow-up visit) and comprised Primary Efficacy Subset, while one patient was lost at the first week of treatment. The disposition of patients and distribution between the study groups are shown in the flow chart in Figure 5. The schedule of examinations and procedures is in Table 3.

The study was conducted at the National Family Medicine Training Center, Tbilisi, Georgia, with the approval of the Biomedical Research Ethics Committee of Tbilisi State Medical University and National Council on Bioethics (Registration Nr 2-2021/86, date of final protocol approval 19 February 2021). ClinicalTrials.gov Identifier: NCT04795557. https://www.clinicaltrials.gov/ct2/show/NCT04795557 (accessed on 14 February 2022).

All participants provided written informed consent to join the study before inclusion. The information about the study was presented to the study participants in Georgian and English languages by local regulations. The patient information sheet described the study procedures, the aims, expected benefits, and potential risks. The principal investigator explained the document’s content to each patient in detail. Before signing the informed consent form, the patients had time to consider the information to confirm that they understood it and were willing to participate in the study.

### 4.2. Study Population

#### 4.2.1. Recruitment and Screening

Individuals were recruited by doctors National Family Medicine Training Center in Tbilisi, Georgia, to attend patients to the clinics. The screening procedures for eligibility to take part in the study were conducted after receiving written informed consent. All relevant principles of the declaration of Helsinki, the ICH guidelines, and EMEA clinical trials guidelines were considered. In the course of the patient’s initial visit to the study site, exclusion and inclusion criteria were checked against the eligibility checklist, and individuals interested in participating received relevant information about the study. Patients were evaluated by a physical examination performed by an investigator, and appropriate lab tests were conducted. Patients consented to the study and underwent randomization when inclusion criteria were met.

#### 4.2.2. Inclusion and Exclusion Criteria

The population for this study consisted of COVID-19 patients, aged 21 to 72 years (mean age: 48.85 ± 13.86 years), with confirmed diagnosis based on positive SARS-CoV-2 test and at least three Long COVID-19 symptoms including fatigue, headache, respiration problems (dyspnea or polypnea), sweat, cognitive disorders (attention, memory, anxiety, and depression), loss of smell (anosmia) and taste (ageusia), cough, pain in joints, muscles and chest, for the last 30 days before recruitment for the study, and having been discharged from a COVID Hotel, Online Clinic isolation, or hospital/Intensive Care Unit (ICU) admission. Subjects were under observation or admitted to a controlled facility or hospital and able to take medication alone and to give informed consent. They had from mild to severe HAM-A score (7–30, mean value: 16.5 ± 4.9), from mild to severe HADS score (7–29, mean value: 15.6 ± 4.5), and overall normal levels of blood serum cytokines IL-6 (mean 6.1 pg/mL, normal level < 30), D-dimer (mean 119 pg/L, normal level < 250), C-reactive protein (mean 7.9 mg/L, normal level < 350), and creatinine (mean 76 μM, normal level in female 52–92, male 65–120), as estimated at the baseline (visit 1), Table 3.

Post-COVID-19 patients discharged from COVID Hotel isolation, Online Clinics, or hospital/Intensive Care Unit (ICU) with Long COVID symptoms longer than three months were excluded. Patients admitted with the severe acute respiratory syndrome and diagnosed with an etiologic agent other than SARS-CoV-2 and a patient under invasive mechanical ventilation were also excluded from the study. Other exclusion criteria were as follows: renal failure requiring dialysis or creatinine ≥ 2.0 mg/dL, tube feeding or parenteral nutrition, respiratory decompensation requiring mechanical ventilation, uncontrolled diabetes type 2, autoimmune disease, pregnant or lactating women, taking antibiotics for a reason other than COVID-19 at enrollment, chronically weakened immune system (AIDS, lymphoma, corticosteroid therapy, immunosuppressive pathology), treated with chemo-radio-corticosteroid treatment in the last six months, active cancer, taking immunosuppressive drugs, participating in another clinical trial, or any other condition that would prevent safe participation in the study. The patients did not receive any medication to avoid influencing the outcome measures during the clinical trial.

#### 4.2.3. Participant Withdrawal

Participants were free to withdraw from the study at any time without giving a reason and with no adverse consequences. There were 22 cases of participant dropouts from the study.

#### 4.2.4. Data Sets Analyzed

All enrolled and randomly allocated to treatment patients were included in the intention to treat analysis. Efficacy subset analysis per protocol (P.P.) was performed for the subset of patients with Long COVID symptoms at the baseline (visit 1) and when they completed the study therapy. A per-protocol (P.P.) analysis aims to identify a treatment effect on the symptoms. Therefore, some patients (from the complete analysis set) needed to be excluded from the population used for the P.P. analysis (P.P. population) (Figure 5).

### 4.3. Intervention and Comparator

ADAPT-232/Chisan^®^ oral suspension (Product Registration No: 2006–2004 at Swedish MDA, Appendix A) was manufactured according to ICH Guidelines for GMP (Swedish Herbal Institute AB, Vallberga, Sweden). One daily dose (60 mL oral solution) contained: 180 mg of soft extract of *Rhodiola rosea* L., radix et rhizome (DER 2.5–5.0:1, extractions solvents —70% ethanol and water) corresponding to 0.45–0.90 g of dried plant material, 600 mg of soft extract of *Schisandra chinensis* (Turcz.) Baill., fructus, (DER 2.0–5.0:1, extraction solvent—95% ethanol) corresponding to 1.2–3.0 g of dried plant material, 156 mg of soft extract of Eleutherococcus senticosus (Rupr. et Maxim) Maxim, radix, (DER 17–30:1, extraction solvents—70% ethanol and water) corresponding to 2.64–4.68 g of dried plant material, and inactive ingredients: dark syrup, anhydrous ethanol, glycerol, caramel aroma, polysorbate 80, methyl parahydroxybenzoate (E218), anhydrous citric acid, rosemary extract, propyl parahydroxybenzoate (E216), ginger extract, and potassium sorbate and water. Placebo suspension containing the same inactive ingredients had a similar appearance, smell, and color and was organoleptically indistinguishable from serum-containing active pharmaceutical ingredients. The products were packaged, blinded, and labeled with the product name, study code, and storage conditions. The investigational products (IPs) were bottled, labeled, and packed by the Swedish Herbal Institute AB, Sweden, according to national requirements regarding the use for clinical trial investigation. The label also included the drug name, study code, and storage conditions. Placebo and ADAPT-232 bottles were provided to patients in boxes containing two 500 mL bottles (the amount required for the treatment period of 14 days is 840 mL). Labels were identical but did not contain an individual identifier (the code). Each patient received two bottles on visit 1 (Day 1). Participant’s ID—the name/initials of the patient allocated to treatment code number—were handwritten by the investigator.

Herbal preparation was qualitatively and quantitatively tested by HPLC [17] according to product specifications. All analytical methods were validated for selectivity, accuracy, and precision. Reference samples were retained and stored at the Swedish Herbal Institute AB (Vallberga, 31250 Sweden).

All study products were maintained in a secure place under appropriate storage conditions. The storage was locked and only available to authorized personnel. The investigational product label and investigator brochure specified the proper storage and shipment conditions.

#### 4.3.1. Doses and Treatment Regimens

The preparations were orally administered in the daily dose of 60 mL for 14 consecutive days. An amount of 30 mL of oral solutions was taken in the morning after breakfast and evening after dinner. The investigator was responsible for maintaining the study products. Drug accountability for this study was carried out in accordance with the standard procedures.

The treatment regime was quadruple-blinded, and hence neither the participants, care provider, investigators, nor outcomes assessor knew which patients were given the placebo or ADAPT-232.

#### 4.3.2. Randomization and Blinding

Study preparations were labeled by a qualified pharmacist (QP) at the manufacturing site using a randomization sequence generated by PRIZM GraphPad software (2017 Online version) "Random number generator" (https://www.graphpad.com/quickcalcs/randomize1.cfm, accessed on 14 February 2022). The randomization sequence table contained two columns (A and B) filled with randomly distributed unique numbers from 1 to 100. It included information on the content of each bottle and package—how placebo and serum packages/containers were encoded. The assignment of serum and placebo to A and B groups/sets of packages to treatment code was encoded by a qualified pharmacist (QP) during the study medication (treatment) randomization procedure at the manufacturing site. The QP kept the randomization sequence and treatment randomization code at the product manufacturing site (the sponsor) until the study was finalized.

#### 4.3.3. Allocation Concealment

All the bottles have allocation concealed random numbers printed on the label. The randomization list with the PI was blinded, so the PI administered the bottles sequentially as per the list. The random sequence of the treatments was kept confidential by the QP at the manufacturing site until the study was finalized. It was provided to the Principal Investigator before statistical evaluation of the results when all the patients had completed the treatment.

#### 4.3.4. Implementation and Blinding

At the first visit, all participants received a consecutive number starting from 001 to 100. They were identified with a unique number according to the randomization sequence. Patients were sequentially enrolled by the Principal Investigator and assigned to a random number, and received the bottles in the corresponding package. The investigator generated the participant list and gave the treatment code no. (from 1 to 100) to every patient. He wrote the patients’ names in Case Report Forms (CRF) and put them on the labels of the packages. The table shows the names of patients and corresponding study medication numbers (treatment code no. mentioned on the label of packages) provided to the participant.

Blinding for trial subjects was performed using labeled packages containing bottles with liquid of identical appearance. Study medication was delivered to the clinic pre-labeled and coded according to the randomization list. The randomization code was kept secret from the clinic and the participating investigators until the code was broken after finalizing the study. In this way, the study medication was also blinded to the investigators, and the study was double-blind.

Thus, the study participants’ list, identifying the patients and the study medication packages (numbers) used in the study, were kept by the Principal Investigator. They were provided for statistical analysis at the end of the study together with the treatment code numbers received from the QP. The treatment code providing the information about the actual assignment of groups A and B to ADAPT-232 and placebo were broken by QP, and after that, statistical analysis of the datasets was completed, and the results of the study were obtained.

#### 4.3.5. Evaluation of Compliance

Participants were asked to take their daily dose of 60 mL oral solution. Participants were questioned about their overall compliance with the study protocol upon their visits, and the study personnel measured the remaining liquid. The doctor monitored compliance. He checked patients’ records in a unique form attached to the Case Report Form. The doctor checked overall compliance with the study protocol upon their visits, and a measuring cylinder measured the remaining liquid at the end of the study.

### 4.4. Efficacy and Safety Outcomes and Endpoints

The primary efficacy outcome measures of the study were:Duration of symptoms of Long COVID: time (days) from randomization to when symptoms disappear. Time frame: change from baseline during the treatment period and follow-up (from Day 1 to Day 14 and Day 21 after randomization).Number of participants clinically recovered: number of participants without symptoms of Long COVID. Time frame: change from baseline during the treatment period and follow-up (from Day 1 to Day 14 and Day 21 after randomization).Length of homestay/sick-listed: time (days) at home/sick-listed. Time frame: change from baseline during the treatment period and follow-up (from Day 1 to Day 14 and Day 21 after randomization).The severity of the Long COVID symptoms: time from randomization to relief of total and individual Long COVID symptoms. Patients were assessed for changes in clinical signs: headache, fatigue, physical activity, depression and anxiety, anosmia, ageusia, hair loss, cough, fever, sweat, pain in muscles, chest, and joints. The medians and hazard ratio were measured and compared between groups. Time frame: change from baseline during the treatment period and follow-up (through 21 days after randomization).Physical activity and daily workout: assessed by Habitual Physical Activity Questionnaire Score and duration of walking (min). Time frame: change from baseline during the treatment period and follow-up (from Day 1 to Day 14 and day 21 after randomization).Cognitive performance score: d2 test of attention and memory. Time frame: change from baseline during the period of the treatment and follow-up (from Day 1 to Day 14 and Day 21 after randomization).The severity of anxiety and depression was assessed by Hospital Anxiety and Depression Scale (HADS). Time frame: change from baseline during the treatment period and follow-up (from Day 1 to Day 14 and Day 21 after randomization).The severity of anxiety assessed by Hamilton Anxiety Rating Scale. Time frame: change from baseline during the treatment period and follow-up (from Day 1 to Day 14 and Day 21 after randomization).Hypercoagulation marker: D-dimer, concentration in the serum, pg/L. Time frame: change from baseline during the period of the treatment and follow-up (from Day 1 to Day 14 and Day 21 after randomization).Immune response marker: IL-6 concentration in the serum, pg/mL. Time frame: change from baseline during the treatment period and follow-up (from Day 1 to Day 14 and day 21 after randomization).Inflammatory marker: C-reactive protein in serum, mg/L. Time Frame: change from baseline during the period of the treatment and follow-up (from Day 1 to Day 14 and Day 21 after randomization).Inflammatory marker: creatinine in serum, μM. Time frame: change from baseline during the treatment period and follow-up (from Day 1 to Day 14 and Day 21 after randomization).

The efficacy endpoints of the study were the differences in the relief of inflammatory symptoms in ADAPT-232 and placebo groups of patients. The outcomes include changes in the severity of inflammatory symptoms, measured using various scales’ scores, from the baseline to the end of therapy (Day 14) and follow-up period (21 days).

Safety and tolerability were evaluated by monitoring the frequency, duration, and severity of adverse events.

### 4.5. Statistical Analysis

The clinical data recorded were recorded in standardized case report forms. These data were tabulated in an Excel dataset (Appendix A) that was used in statistical analysis using Prism software (version 3.03 for Windows; GraphPad, San Diego, CA, USA).

Statistical analysis was performed using "observed" data for time-to-event outcomes of the intent-to-treat (ITT) population, defined as all randomly assigned patients who received at least one dose of the study medication.

The mean outcomes were compared at baseline for patients who received ADAPT-232 vs. placebo by the Student’s parametric independent-measures t-test (variables with normal distribution) or Mann–Whitney non-parametric test, depending on results of the normality test. Baseline characteristics were also compared using tools for assessing column statistics and the KW non-parametric one-way ANOVA rank order test with post hoc Dunn’s Multiple Comparison Test to compare four groups. The same statistical tools were applied for the analysis of changes within treatment groups over time (ITT analysis). Within-group repeated measures analysis of variables was conducted with one-way, repeated measures ANOVA (data with normal distribution) or the Friedman non-parametric rank test. Additional statistical analysis was performed for patients who had Long COVID symptoms on the day of randomization (visit 1), completed the treatment, and passed all tests in all visits to the clinic.

Assessment of the efficacy of study medications was achieved by comparison of mean changes from the baseline (differences before and after treatment of every single patient) between groups using two-way between–within ANOVA in which an interaction effect indicates a different response over time between the two groups and would therefore signal a treatment effect, as well as by multiple comparison t-test (one unpaired test per row).

The level of statistical significance was set at 5% in all methods.

For the duration of the symptoms, Kaplan–Meier curves were generated for all endpoints, and medians were calculated from those curves. The treatment arms were compared by Gehan–Breslow–Wilcoxon and Mantel–Cox log-rank tests depending on the results of the normality test. The estimates of treatment hazard ratios based on log-rank tests and 95% CIs were calculated.

#### Sample Size Considerations

The sample size was calculated based on previous studies [11,12,13] using the formula *n* = 16σ2/W2, where *n* is the minimal sample size, σ the variance and W the confidence interval size. This method suggested a minimal sample size of 28. The necessary sample size was estimated at 80, i.e., 40 subjects per arm. Assuming a 20% dropout rate, which would increase our intended sample size, 50 participants in each group would be enough to detect a significant difference between placebo and serum groups. The proposed sample size (*n =* 100, or 50 per treatment condition) has 90% power to detect an effect size of 0.74 and 80% power to detect an effect size of 0.85, using a 2-group t-test with a 0.05 two-sided significance level. The estimated effect sizes from this study can then be used in a large-scale study of Chisan^®^/ADAPT-232 in Long COVID-19 that is intended as a follow-up to this pilot study.

## 5. Conclusions

Overall, this pilot study demonstrates that Chisan^®^/ADAPT-232 can increase physical performance in Long COVID. It also suggests that Chisan^®^/ADAPT-232 might be useful for preventing the progression of renal failure associated with increasing creatinine. Essentially, this study is the first randomized placebo-controlled clinical trial of the efficacy of a pharmaceutical in Long COVID.

## Figures and Tables

**Figure 1 pharmaceuticals-15-00345-f001:**
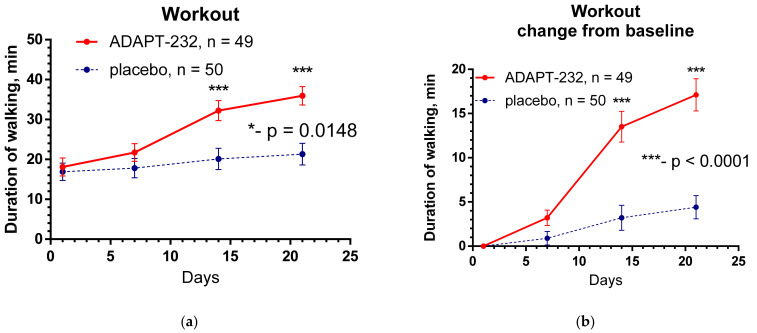
(**a**) Workout time (mean ± SEM) of patients in group A (ADAPT-232) and group B (placebo) over the time from Day 1 to Day 21. The changes from the baseline within groups A and B over time were significant (*p* < 0.0001; calculated by repeated measures ANOVA); two-way ANOVA estimated the significant interaction between treatment groups over time; *—*p =* 0.0148. (**b**) Between-groups comparison of the changes of workout from the baseline over time shows significant interaction (*p* < 0.0001) and very significant difference (*p* < 0.0001) between groups A and B. The ADAPT-232 treatment significantly increases patients’ workouts compared to placebo. ***—*p*< 0.001. For details of statistical analysis, see Appendix A.

**Figure 2 pharmaceuticals-15-00345-f002:**
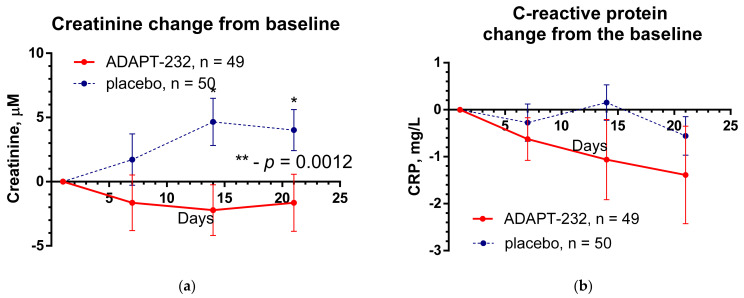
The changes from the baseline of the levels (mean ± SEM) of (**a**) creatinine and (**b**) C-reactive protein in the blood of patients in group A (ADAPT-232) and group B (placebo) over the time from Day 1 to Day 21. Between-groups comparison of the creatinine shows a significant difference (*p* = 0.0012) between groups A and B. The ADAPT-232 treatment significantly decreased blood creatinine compared to placebo. For details of statistical analysis, see Appendix A. Between-groups comparison of the changes of C-reactive protein in blood from the baseline over time shows no interaction (*p* = 0.7100) and no significant difference (*p* = 0.1276) between groups A and B. The ADAPT-232 treatment has no statistically significant effect on C-reactive protein level in blood compared to placebo. *—*p* < 0.05 and **—*p*< 0.01. For details of statistical analysis, see Appendix A.

**Figure 3 pharmaceuticals-15-00345-f003:**
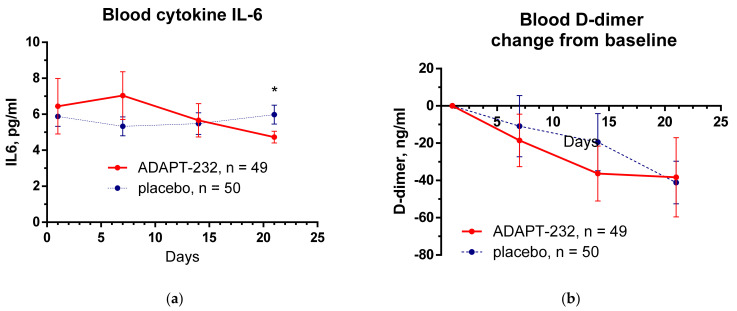
The changes from the baseline of the levels (mean ± SEM) of (**a**) cytokine IL-6 and (**b**) D-dimer in the blood of patients in group A (ADAPT-232) and group B (placebo) over the time from Day 1 to Day 21. Between-groups comparison of the changes of cytokine IL-6 level in the blood from the baseline over time shows no interaction (*p =* 0.4369) and no significant difference (*p* = 0.5879) between groups A and B. The ADAPT-232 treatment has no statistically significant effect on cytokine IL-6 in blood compared to placebo. Between-groups comparison of the changes of blood D-dimer level from the baseline over time shows no interaction (*p* = 0.8920) and no significant difference (*p* = 0.5782) between groups A and B. The ADAPT-232 treatment has no statistically significant effect on blood D-dimer compared to placebo. *—*p* < 0.05. For details of statistical analysis, see Appendix A.

**Figure 4 pharmaceuticals-15-00345-f004:**
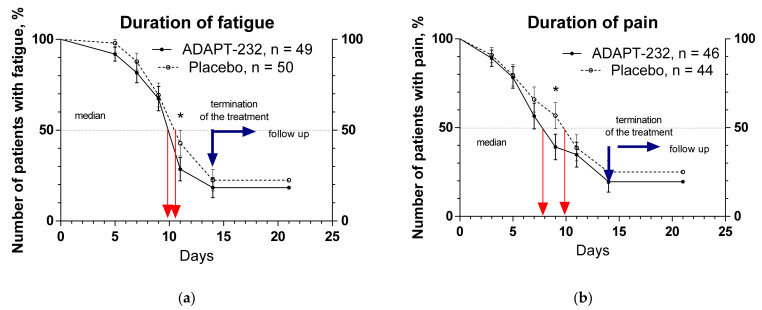
Kaplan–Meier curves show the decrease of the duration of fatigue and pain over the time from randomization (Day 1) to the end of the treatment (Day 14) and followed up for one week (Day 21) and the number of patients who experienced these symptoms of Long COVID: (**a**) fatigue, (**b**) pain. *—*p* < 0.05.

**Figure 5 pharmaceuticals-15-00345-f005:**
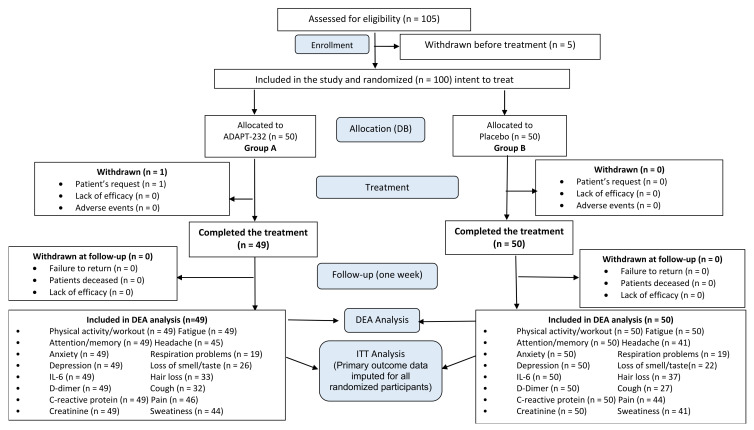
Schematic diagram of the trial: disposition of patients.

**Table 1 pharmaceuticals-15-00345-t001:** Baseline demographic characteristics, study outcome measures, and laboratory hematological and biochemical measurements.

	Unit	Group A ADAPT-232, *n =* 50		Group B Placebo *n =* 50		Signif. of Difference
Parameters		Mean	S.D.	Mean	S.D.	*p*-Value
Age	years	**49.52**	12.57	**48.18**	15.13	0.6312 ^a^
Gender	Male/Female	**7/42**		**7/43**		
BMI	kg/m^2^	**26.93**	4.054	**26.62**	4.750	0.7196 ^a^
Start of symptoms	weeks	**2.720**	0.6402	**2.760**	0.6247	0.7525 ^b^
Compliance	%	**101.0**	1.998	**100.8**	2.437	0.1804 ^b^
Fatigue	A.U.	**1.694**	0.585	**1.780**	0.648	0.489 ^a^
Headache	A.U.	**1.184**	0.565	**1.100**	0.707	0.518 ^a^
Respiration problems	A.U.	**0.449**	0.6145	**0.4600**	0.7060	0.876 ^b^
Organoleptic disorders	A.U.	**0.5600**	0.7602	**0.4400**	0.5014	0.778 ^b^
Hair loss	A.U.	**0.8660**	0.1237	**0.8953**	0.1266	0.4996 ^a^
Body temperature increase	C	**<37**		**<37**		>0.05
Cough	AU	**0.7347**	0.6047	**0.6000**	0.6061	0.2711 ^a^
Pain in muscles, chest, and joints	A.U.	**1.245**	0.5962	**1.160**	0.6503	0.5002 ^a^
Sweatiness	A.U.	**1.204**	0.7065	**1.120**	0.7730	0.6417 ^b^
Stay at home/sick-listed	days	**0.3265**	0.4738	**0.8000**	2.356	0.6866 ^b^
Physical activity	A.U.	**16.43**	2.850	**15.56**	4.305	0.3702 ^b^
Physical activity (daily walk)	min	**5.534**	0.0628	**5.162**	0.0757	0.7004 ^a^
Decreased attention (d2 test)	%E (errors)	**25.51**	16.70	**26.43**	18.93	0.9849 ^b^
Anxiety (mild 14–17; moderate 18–24; severe > 25)	HAM-A score	**16.46**	5.388	**16.44**	4.634	0.8674 ^b^
Depression (mild 8–10, moderate 11–14, severe > 15)	HADS score	**15.30**	4.595	**15.78**	4.473	0.5978 ^a^
Blood serum cytokines IL-6 (normal level < 5.186)	pg/mL	**6.443**	10.95	**5.883**	3.919	0.3763 ^b^
D-dimer (normal level < 250)	pg/L	**133.3**	195.8	**105.2**	99.46	0.9492 ^b^
C-reactive protein (normal level < 350)	mg/L	**5.578**	8.313	**10.35**	18.74	0.6630 ^b^
Creatinine (female 52–92, male 65–120)	μM	**77.18**	11.50	**75.50**	10.69	0.5557 ^b^
Total Leukocyte count, WBC	10^3^ u/L	**8.96**	12.11	**7.55**	4.42	0.44
Erythrocytes, RBC	10^6^ u/L	**4.57**	0.39	**4.63**	0.46	0.51
Hemoglobin, Hb	g/L	**128.26**	11.77	**130.84**	11.39	0.27
Hematocrit, HCT	%	**38.53**	3.30	**39.12**	3.46	0.39
Platelet Count	10^3^ u/L	**240.16**	57.65	**251.86**	59.64	0.32
Absolute Neutrophil count	10^3^ u/L	**60.89**	8.30	**60.49**	8.60	0.81
Total Lymphocyte count	10^3^ u/L	**31.13**	7.75	**30.03**	9.33	0.52
Monocyte count	10^3^ u/L	**2.79**	3.43	**3.05**	3.40	0.70
Eosinophil count	10^3^ u/L	**4.36**	1.61	**4.22**	1.57	0.65
Basophil count	10^3^ u/L	**0.09**	0.24	**0.07**	0.22	0.75
Erythrocyte sedimentation rate, ESR	mm/h	**19.56**	11.18	**15.94**	9.53	0.08

**Table 2 pharmaceuticals-15-00345-t002:** Duration (days) of symptoms from the day of randomization and the significance of the difference between groups A (ADAPT-232) and B (placebo).

	Group A ADAPT			Group B Placebo			Significance of Difference
Parameters	Mean	SD	*n*	Mean	SD	*n*	*p* Value
Fatigue, days	**11.96**	4.899	49	**12.98**	0.648	50	0.2662 ^b^
Headache, days	**10.40**	5.336	45	**9.244**	4.700	41	0.3582 ^b^
Respiration problems, days	**6.579**	3.485	19	**6.895**	5.259	19	0.7619 ^b^
Organoleptic disfunctions, days	**11.58**	6.543	26	**10.64**	5.728	22	0.8383 ^b^
Hair loss, days	**16.70**	5.823	33	**15.41**	5.231	37	0.3317 ^a^
Cough, days	**8.094**	4.855	32	**6.259**	4.809	27	0.0219 ^b^
Pain in muscles, chest and joints, days	**10.67**	6.056	46	**11.77**	6.243	44	0.3668 ^b^
Sweatiness, days	**12.07**	6.211	44	**12.24**	6.102	41	0.7605 ^b^
**Average duration of all symptoms, days**	**7.640**	**2.724**	**50**	**7.463**	2.469	50	0.5375 ^b^
Stay at home/sick-listed, days	**5.918**	9.089	49	**5.940**	9.224	50	0.9839 ^b^

^a^—parametric unpaired *t*-test; ^b^—non-parametric Mann–Whitney test.

**Table 3 pharmaceuticals-15-00345-t003:** Schedule of examinations and procedures.

	Treatment							Follow-Up
	Day 1 Screening	Day 3	Day 5	Day 7	Day 9	Day 11	Day 14	Day 21
Doctor’s visits	1 Baseline			2			3	4
Eligibility check/Information	*							
Informed consent	*							
Clinical examination	*			*			*	*
Enrollment and allocation to intervention	*							
Treatment (Kan Jang and placebo)	*	*	*	*	*	*	*	
*Biomarker assessments*								
Body temperature (fever)	*	*	*	*	*	*	*	*
COVID-19 PCR test	*						*	
Blood serum cytokine IL-6 (pg/mL)	*			*			*	*
D-dimer (pg/L)	*			*			*	*
C-reactive protein (mg/L)	*			*			*	*
Creatinine μM	*			*			*	
*Clinician and observer reported outcomes assessments*								
Cognitive performance (tests forattention and memory): d2 test	*			*			*	*
Tests for anxiety/depression: HADS Scale * HAM-A Scale **	*			*			*	*
Drug intake accountability							*	
Adverse events				*			*	*
*Patient-reported outcomes assessments*								
Long COVID symptoms: Fatigue Headache Loss of smell and taste Hair loss Difficult and rapid respiration Cough Pain in joints, muscles and chest Sweatiness	*	*	*	*	*	*	*	*
Workout, min	*			*			*	*
Physical activity (questionnaire)	*			*			*	*

Anxiety and depression subscales. * HADS: normal 0–7, mild 8–10, moderate 11–14, and severe 15–21; ** HAM-A: 14–17 = mild anxiety, 18–24 = moderate anxiety, 25–30 = severe anxiety.

## Data Availability

Data is contained within the article and Appendix A.

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
