# Peer review of "Efficacy of Adaptogens in Patients with Long COVID-19: A Randomized, Quadruple-Blind, Placebo-Controlled Trial"

_pharmaceuticals, 2022, doi:10.3390/ph15030345_

Round 1
Reviewer 1 Report
Dear Authors I have read the manuscript and I think that it is very interesting, however I think that more data related to cytokines must be added in both results and discussion.
In fact the role of IL-6 has been well documented and some authors reported that other proinflammatory mediators are involved in neuroinflammation. Therefore I think that you must add these data in results as well as in discussion (see doi: 10.1371/journal.pone.0251819).
Moreover, please add the CT of COVID for each group in order to evaluate if the viral activity could play a role in clinical response
Reviewer 2 Report
Minor revision
The descriptive analysis in the Result is not particularly compelling. As a matter of fact, it is necessary to emphasize the Result.
In my opinion and observation, the outcome portion of this study requires a much more conventional description. As we all know, a lot of studies on COVID-related research have already been published. As a consequence, it is necessary to emphasise the outcomes and compare them to the previous report.
